# Potential Variables for Improved Reproducibility of Neuronal Cell Grafts at Stroke Sites

**DOI:** 10.3390/cells11101656

**Published:** 2022-05-17

**Authors:** Joanna Krzyspiak, Kamran Khodakhah, Jean M. Hébert

**Affiliations:** 1Department of Neuroscience, Albert Einstein College of Medicine, Bronx, New York, NY 10461, USA; joanna.e.krzyspiak@gmail.com (J.K.); k.khodakhah@einsteinmed.edu (K.K.); 2Stem Cell Institute, Albert Einstein College of Medicine, Bronx, New York, NY 10461, USA; 3Department of Genetics, Albert Einstein College of Medicine, Bronx, New York, NY 10461, USA

**Keywords:** ischemia, neocortex, transplantation, vascular endothelial cells, blood vessel

## Abstract

Interest is growing in using cell replacements to repair the damage caused by an ischemic stroke. Yet, the usefulness of cell transplants can be limited by the variability observed in their successful engraftment. For example, we recently showed that, although the inclusion of donor-derived vascular cells was necessary for the formation of large grafts (up to 15 mm^3^) at stroke sites in mice, the size of the grafts overall remained highly variable. Such variability can be due to differences in the cells used for transplantation or the host environment. Here, as possible factors affecting engraftment, we test host sex, host age, the extent of ischemic damage, time of transplant after ischemia, minor differences in donor cell maturity, and cell viability at the time of transplantation. We find that graft size at stroke sites correlates with the size of ischemic damage, host sex (females having graft sizes that correlate with damage), donor cell maturity, and host age, but not with the time of transplant after stroke. A general linear model revealed that graft size is best predicted by stroke severity combined with donor cell maturity. These findings can serve as a guide to improving the reproducibility of cell-based repair therapies.

## 1. Introduction

In 2019, there were over 100 million prevalent cases of stroke worldwide [1]. In the event of a stroke, neurons die and are not replaced, often resulting in permanent functional deficits. Currently, the only treatments for stroke are endovascular thrombectomy and the injection of a thrombolytic to reperfuse the ischemic tissue [2] in an attempt to preserve the penumbra, a region of potentially salvageable tissue that surrounds an ischemic, necrotic core. However, there is as yet no therapy that leads to a significant recovery in cases of severe loss of function.

In an attempt to improve neurological function, stem cell transplantation therapy has shown promise in improving functional outcomes in clinical trials with minimal adverse events [3,4]. However, the benefits of transplantation thus far, rather than due to tissue replacement, are due instead to the bystander effect, whereby factors secreted by the transplanted cells benefit surrounding tissue [5,6,7]. Therefore, there remains a need to develop methods to replace injured or diseased tissue to restore lost function, particularly for the neocortex, the site of our higher-order thinking, and a common location for injury such as stroke.

Studies in mice have shown that transplanted neocortical precursor cells can differentiate into proper neuronal cell types, a project to appropriate brain targets, synaptically connect with host neurons, and, for example, normally respond to sensory input or elicit motor output (e.g., [8,9,10]). These studies support the possibility of functionally replacing damaged neocortical tissue. However, cell transplantation in stroke patients is still in its infancy, as the number of total participants remains small, and the exact conditions needed for proper integration and safety are still under investigation [6]. For example, a challenge facing the field is getting transplanted cells, which are usually purified cell populations, to survive well in the host. It has been reported that up to 80% of transplanted cells die within three days of injection [11]), making the obstacle of improvement in cell viability a priority to resolve. Our previous work has shown that the addition of vascular precursor cells to the donor transplant population improves the survival of grafts [12]. Many additional challenges remain, such as recapitulating the cell-type composition, cytoarchitecture, and connectivity of native neocortical tissue within transplant-derived tissue. In addition, there is considerable variability in transplant sizes, even when transplants include supporting cell types such as vascular and glial cells (e.g., [12,13,14]). The cause of this variability remains largely unclear.

Here, we examined variables that might affect graft outcomes. We explored a number of factors such as age and sex of the host, ischemic stroke severity, timing of transplant after stroke, embryonic stage of donor cells, and cell viability at the time of transplantation. The extent of ischemic stroke damage, sex and age of the host, and embryonic stage of the donors showed significant effects on the resulting size of the grafts. Our findings could help guide future experiments to determine what cellular and host conditions are optimal for transplant integration and survival.

## 2. Materials and Methods

Animals. Animal handling followed an approved protocol by the Albert Einstein College of Medicine Institutional Animal Care and Use Committee in accordance with National Institute of Health guidelines. A total of 59 CD1 adult male and female mice of 2–5 months of age were used as hosts in this study. The number of mice used in each experimental group is indicated in the figures. Transgenic donor embryos were generated from mice of mixed backgrounds. For experiments analyzing donor vascular cell contribution, males carrying *Mesp1^Cre^*^/+^ were crossed to females homozygous for *Rosa26CAG^lox-stop-lox-eGFP^* or *Rosa26^lox-stop-lox-tdTomato^* [15,16]. Mice were housed on a 12 h light-dark cycle with a maximum of five animals in a cage and provided with water and chow ad libitum.

Induction of Distal Middle Cerebral Artery Occlusion (dMCAO). Mice were anesthetized under 1.5% isoflurane, and after a small incision was made in the skin and a skin flap folded back, a small hole was drilled in the skull over the distal left middle cerebral artery (MCA). Permanent cortical ischemia was induced by cauterization and disconnection of the MCA at a location in the distal trunk before splitting into the anterior and posterior branches. The skin was then sutured closed. The extent of stroke (or ischemic damage) was measured as the lost fraction of cortical volume in the stroke hemisphere of coronal sections, normalized to the cortical volume of the contralateral hemisphere; volume (area by thickness of sections) of ischemic tissue loss was extrapolated from quantifications of every fourth section through the entire ischemic area.

Preparation of Donor Cells from Mouse Embryos and Transplantation. Embryos of the desired genotype were harvested from pregnant females at ~E12.5. The developing cortex was collected with overlying primitive leptomeninges (arachnoid and pia mater). At this stage, the cortical primordium is comprised of neuroepithelial precursors, early born neurons, vascular cells, and microglial cells [17]. Cortices were incubated in accutase for 20 min to ensure dissociation of cells. The single-cell nature of the donor cells was confirmed under light-field microscopy. After spinning, cells were resuspended at 50,000 cells/µL in a medium containing Neurobasal, N2, B27, penicillin-streptomycin, GlutaMax, and HEPES buffer. Host mice that had undergone dMCAO 1, 3, and 7 days prior were anesthetized with 1.5% isoflurane and injected with 2 µL of cell suspension bilaterally at −0.5 mm posterior and +/− 3.3 mm mediolateral to bregma, at a depth of 2 mm from the surface of the skull, which corresponds to the primary somatosensory cortex. Cell suspensions were injected using a 30-gauge Hamilton syringe at a rate of 0.150 µL/min. The needle was left in for 5 min before slowly retracting to minimize backflow.

Immunofluorescence. Mice were perfused with 4% paraformaldehyde (PFA) intracardially 2 weeks after transplantation and postfixed overnight in 4% PFA at 4 °C. Brains were cryoprotected with 30% sucrose for 48 h, embedded in OCT, and stored at −80 °C. Sections were cut at 30 µm and immunostained as floating sections. Where needed, antigen retrieval was performed by heating sections for 45 min at 90 °C in 10 mM sodium citrate pH 6.0. The following primary antibodies were used: CD105 (1:100) Cat No. 120402 BioLegend, PECAM (1:100) Cat No. ab28364 AbCam, GFAP (1:500) Cat No. 13-0300 ThermoFisher, GFP (1:250) Cat No. A-11122 ThermoFisher, Satb2 (1:500) Cat No. ab92446 AbCam, Ctip2 (1:500) Cat No. ab18465 AbCam, Ki67 (1:500) Cat No. ab15580 AbCam, Iba1 (1:500) Cat No. 019-19741 Wako, and Fos (1:500) Cat No. 226 004 Synaptic System. Primary antibodies were incubated overnight at 4 °C and in Alexa-488, Alexa-568, and Alexa-647 secondaries (1:500 Invitrogen) for 1 h at room temperature.

Image Analysis of Transplants. Blood vessels were quantified using the software AngioTool [18]. The area of donor-derived blood vessels was divided by the area of total blood vessels to determine the fraction of vessels stemming from the donor population. The size of transplants was quantified using ZEN software (Blue edition, Zeiss). The area of the transplant was measured in one of every four sections throughout the transplanted tissue, and the volume was estimated by multiplying the area by the thickness of the section times four.

Statistical Methods. Statistical analyses were performed using GraphPad Prism. Data were analyzed for statistical significance using the Wilcoxon Ranked test for pairwise analyses of grafts and the Mann–Whitney U test for comparisons between two groups. Sample sizes and *p* values for each experiment are listed in the figure legends. Spearman correlation coefficients were used to determine relationships between variables. Multiple linear regression was applied to examine the correlation between graft size, the embryonic stage of donor cells, and stroke severity.

## 3. Results

### 3.1. Embryonic Stage of the Donor Cells

Our donor embryos are timed for a target harvest of E12.5. However, because of the differences in the exact time of conception, the age of donor embryos may differ up to 12 h between litters and even several hours within litters (Figure 1A) [19]. These differences may be significant depending on the stage of the embryo. At E12.5, 60% of neocortical progenitor cells are actively dividing, and by E13.5, this number drops to 40% [20]. Therefore, a difference of 12 h in donor embryo ages could impact the resulting graft size. To test this possibility, embryos were imaged and staged according to a 4D morphological atlas [19]. Transplants of neocortical precursors isolated from embryos closer to E12.4 in age resulted in slightly larger grafts after 2 weeks than those from embryos closer to E13.2, with a correlation coefficient of −0.51 (*p* = 0.0027, N = 32) (Figure 1B). This is consistent with the hypothesis that harvesting donor cells at different timepoints, even within a 12 h window, can lead to variability in transplant size.

### 3.2. Donor Cell Viability at Time of Transplantation

For successful and reproducible engraftment, healthy, viable donor cells are required. Therefore, care is taken that each harvest and injection is performed the same way for each experiment, including incubating cells for enzymatic digestion in accutase for equal periods of time, triturating with comparable force and duration, and continuously assessing cells for viability during the experiment. Before each transplant, cell viability was determined with trypan blue staining and noted so that if cells dropped below 75% viability, they were not used for transplantation. We examined if a simple correlation between cell viability immediately before injection and the transplant size existed, however, no correlation was found for transplantation in either the stroke (R = −0.28, *p* = 0.12) or control (R = −0.09, *p* = 0.12) sites (Figure 2B, n = 30 per side).

To address the possibility of selective viability of precursor types in the donor cell population, we immunostained sections for SATB2 and CTIP2 (neurons), OLIG2 (oligodendrocytes), GFAP (reactive astrocytes), CD105 (endothelial cells), and IBA1 (microglia). Compared with neighboring endogenous tissue, the grafts had a slightly smaller fraction of neurons (*p* = 0.018) and a slightly higher fraction of reactive astrocytes (*p* = 0.029) (Figure 2C). The remaining cell types showed no significant differences.

### 3.3. Effect of a Transplantation Delay of 1, 3, and 7 Days after dMCAO

The brain parenchyma during the days after a stroke is a dynamic environment. By 48 h after ischemia, 60% of neurons in the ischemic area stain for TUNEL, a DNA fragmentation marker, and by 7 days after ischemic stroke, 90% of neurons in the infarct core are positive for TUNEL. This was also shown previously when inflammatory cells begin to invade the stroke region [21].

The endogenous vasculature in the infarct area begins to decline soon after the ischemia and precipitously declines by 72 h, though by 14 days after ischemia, new microvessels are apparent along with collagen IV deposits, which are conducive to angiogenesis. In mice with transient MCAO, endothelial cells begin to proliferate 24 h after ischemia and peak at 3 days as new blood vessels form until 21 days postinjury [22]. Therefore, transplanting cells at different timepoints postischemia could change the contribution of donor versus host blood vessels in the graft and the graft size [12].

We performed transplants at 1, 3, and 7 days poststroke. One day poststroke was chosen because this timepoint was previously shown to be a period before extensive environmental changes occur and neuronal death begins to increase [21]. Within 3 days post dMCAO, blood vessel formation begins to decline; therefore, we reasoned that transplanting at 1 or 3 days poststroke might increase the proportion of host-derived blood vessels, which are being induced to regenerate, whereas transplanting at 7 days after might favor donor-derived blood vessel formation. In fact, the average amount of donor-derived vasculature was higher in the 3 days vs. 1 day poststroke timepoints (*p* = 0.004) and even higher at 7 days versus 1 day, but not significantly different between the 3 and 7-day timepoints on the stroke side (*p* = 0.24) (7 days: 46.1% ± 18.1, n = 12; 3 days: 50.2% ± 13.1, n = 6; 1 day: 54%, n = 7), suggesting that the donor vessels were favored in the transplants in the later stages of recovery after dMCAO. To our surprise, there was also a difference between the donor contribution of vessels on the nonstroke contralateral side for the 3-day compared with the 7-day delay group (*p* = 0.02) (Figure 3A,B). Potential explanations for this difference in the nonstroke hemisphere are discussed below.

One week after an aspiration lesion, inflammation detected in a previous study as an increase in GFAP+ astrocytes and Iba1+ microglia can improve graft survival, neuronal integration, vascularization, and proliferation [23]. The inflammatory environment could explain why grafts are larger at stroke sites compared with the control contralateral side (Figure 3C). However, despite the dynamic poststroke environment, our data did not find any significant differences in graft size between transplant days (7 days: 4.1 mm^3^ ± 3.6, n = 17; 3 days: 3.8 mm^3^ ± 4.8, n = 15; 1 day: 3.9 mm^3^ ± 1.9, n = 13) (Figure 3C).

### 3.4. Sex of the Host Mice

As with many diseases and disorders, there are sex-specific differences in clinical outcomes after ischemic stroke. Whereas men are less likely to survive the initial stroke, women have a poorer long-term quality of life [24]. The reasons for these differences remain unclear. We, therefore, tested if there were sex-specific differences in graft size in mice and in the extent of damage sustained in each sex from dMCAO, which was quantified by measuring the area of the host neocortical tissue on the stroke side and dividing it over the area of the control side to obtain a ratio of the remaining tissue. This value was subtracted from 1 to report the fraction of cortical volume lost because of the ischemic stroke. Interestingly, there was no significant difference in the graft size between female and male mice (Female: 3.69 mm^3^ ± 4.14, n = 23; Male: 2.61 mm^3^ ± 2.48, n = 36; *p* = 0.59), nor was there a difference in stroke severity, (Female: 0.23 ± 0.11; Male: 0.24 ± 0.11) (Figure 4A).

Previous studies support the notion that an injured host environment that is still active in its recovery is favorable for successful transplant engraftment [14,23]; therefore, we tested if the size of the graft could be in part dependent on the infarct size. We found that graft size correlated significantly with stroke severity in both sexes, with female mice showing a stronger correlation (R = 0.75, *p* = 0.0001) compared with male mice (R = 0.37, *p* = 0.03) (Figure 4B).

### 3.5. Age of Hosts

Although previous studies have shown that over 75% of strokes in humans occur after the age of 65 [25], many studies are still being carried out on young animals [24]. Increasingly, however, the differences between young and aged tissue in animal models are appreciated in the context of disorders such as stroke [26]. Therefore, we performed experiments as outlined above, but with 18 to 24-month-old mice. There was no statistically significant difference in graft size between young and aged mice (young n = 45, 3.08 mm^3^ ± 3.39; aged n = 10, 2.8 mm^3^ ± 2.49), nor was there a difference in the severity of the stroke (young: 0.24 ± 0.12; aged: 0.25 ± 0.1) (Figure 4C). We were again interested to know if the size of the grafts had any dependency on age. In young mice, there was a significant correlation with size (R = 0.44, *p* < 0.0024), but not in aged mice (R = −0.30, *p* = 0.42) (Figure 4D). Combining all mice resulted in a correlation of R = 0.41 (*p* < 0.0024, N = 54).

Nevertheless, in older mice, grafts often appeared to have vessels of larger caliber than grafts in young mice. Quantifying the ratio of donor versus host vessels was difficult because of high background staining in the area, likely because of nonspecific binding of secondary antibodies to host mouse IgG that seeped into the surrounding tissue through leaky vessels. Overall, transplantation into the stroke sites of older mice appeared successful, although qualitative differences suggest future studies are necessary to assess vascular and neuronal performance.

### 3.6. General Linear Model of Stroke Severity and Embryonic Stage of Donor Cells

More than one component may be driving the distribution of data described above that, when considering only one factor at a time, appear to have high, unexplained variability. From our analysis, the embryonic stage of donor cells and stroke severity appears to have the greatest effect on graft size, the dependent variable. To determine if considering both elements together would be a better predictor of graft size than either variable alone, we combined them using a generalized linear model. Indeed, this resulted in a stronger predictor of transplant size (R^2^= 0.47, parameter estimate stroke = 9.28, *p* = 0.0013; parameter estimate stage = −4.56, *p* = 0.0011, n = 32) than either stroke severity (R^2^= 0.24, parameter estimate stroke = 9.4, *p* = 0.0046) or embryonic stage alone (R^2^ = 0.24, parameter estimate stage = −4.6, *p* = 0.0042) (Figure 4E). Moreover, the combined GLM revealed that the impact of the stroke severity on the graft size was almost double that of the embryonic stage (slopes of 9.28 vs. −4.56, respectively).

### 3.7. Fos Expression as a Proxy for Neuronal Activity

Although we previously assessed the functionality of our transplant paradigm by measuring the electrophysiological activity and connectivity of graft-derived neurons at stroke sites, the accuracy of these data did not allow for a comparison with grafts at nonstroke sites [12]. Induction of Fos expression is commonly used as a marker for neuronal activity [27]. Therefore, we compared Fos expression in grafts at a stroke and nonstroke sites. Under both conditions, there was extensive Fos expression (Figure 5), consistent with spontaneous neuronal activity in developing neocortical neurons [28,29]. Interestingly, the percentage of neurons (CTIP2+ plus SATB2+) expressing Fos was modestly higher in grafts at the stroke sites than in control sites, which could reflect greater neuronal integration and activity (Figure 5C).

## 4. Discussion

Stem cell transplantations have significant variability in cell survival and graft size, which is an unresolved concern for achieving more reliable, reproducible grafts for clinical applications. It is worth noting that in the current study, we did not examine cell type differentiation and electrophysiological integration of grafts, which are ultimately superior measures than the size in assessing functionality. However, by transplanting the same donor cell source in host mice treated the same way as in the current study, we previously found that cell type composition and electrophysiological integration did not noticeably vary with graft size [12], suggesting that graft size in the current experimental paradigm can serve as a proxy for some level of graft functionality. While a number of variables affect the success of grafts, we considered the most salient factors that could be the mediating variability in grafting after ischemic stroke, including donor cell viability at the time of transplant, the timing of transplantation relative to stroke, developmental stage of donor cells, host sex and age, and severity of the stroke.

The ideal timing of transplants relative to the ischemic event is an active area of debate in the field. Although human trials have shown that there is some functional improvement even with transplants performed years after a stroke, it is unknown if the mechanism is due to the graft itself or due to the effects of the surgery. In animal models, transplanting with a 1-week delay after stroke or injury appears to give superior outcomes compared with no delay [14,23], and transplanting 48 h after stroke is preferable to a 6-week delay [30]. Here, we tested timepoints that might reveal differences related to the changes over time in the dynamic injury response, particularly in the context of vessel formation by donor vascular cells. We found that, although there was no detectable difference in graft sizes when comparing transplants performed 1, 3, and 7 days after stroke, there was a significant increase in the contribution of donor-derived endothelial cells to overall vasculature from the 1- to the 7-day delay timepoints.

A previous study found that starting at 24 h after ischemia, endothelial cell (EC) division significantly increases and then begins to subside after about a week [22]). This is in line with our findings showing that host-derived vessels are the dominant contributors to vascularization of transplants injected 1-day post dMCAO. One day poststroke, endogenous vasculogenesis is still robust and, therefore, could account for why the majority of vessels are host-derived. Interestingly, a previous study found an increase in thrombospondin-1, which has been linked to stopping host angiogenesis, at 3 days post dMCAO, with a further increase at 7 days poststroke before decreasing [22]. These findings show that endogenous angiogenesis begins to slow at 3 days poststroke and even more so at 7 days, suggesting that transplants performed at later timepoints might benefit most from donor-derived ECs to vascularize the graft tissue, consistent with our findings. Additionally, inflammation at 1-week postinjury has been shown to be conducive to proliferation and may promote donor vascular cell proliferation [23].

Surprisingly, there was a modest but significant difference between the donor contribution to blood vessels on the nonstroke (contralateral) side for the 3-day delay timepoint compared with the 7-day delay timepoint (*p* = 0.036). Potential explanations are cross-hemispheric effects from the ischemic stroke side, which can manifest as increases in BrdU+ cells or microglia, peaking on the contralateral side at 4 days poststroke [31], or an increase in expression of hypoxia and stem cell-related genes observed 3 days poststroke [32]. This suggests that the environment in the hemisphere contralateral to the stroke is more conducive to the proliferation of the donor cells.

In addition to the timing of the transplant after the stroke, the viability and maturity of donor cells could, in principle, impact the graft size. We showed that the percent of viable cells in the donor cell suspension did not correlate with graft size, at least for relatively minor variations. Roughly 50% more cells are dividing in the developing mouse neocortex at E12.5 versus E13.5, which led us to predict that even slightly younger donor cells could influence the size of our grafts. In fact, we found an inverse correlation between graft size and the embryonic stage of the donor cells, suggesting that fewer proliferating cells, at least within the developmental window that we examined, did reduce graft size.

Sex and age are known variables in stroke outcome in both human and animal models, but how age and sex relate to transplant models remains underexplored. Not only does the parenchyma and immune response differ with age [26], potentially affecting stroke outcomes, but so does blood [33,34]. One study in mice found comparable engraftment of donor neurons between young mice and mice aged 13 months; however, no study has directly compared the severity of the stroke, nor the size of the resulting transplant, as it relates to the age of the host. We previously showed that the severity of a stroke, as measured by the volume of cortical tissue lost, moderately correlated with graft size [12]. This could be due to the environment of larger strokes being more conducive to the proliferation of cells derived directly from fetal tissue, which are suited to survive a hypoxic environment. Remarkably, here we found that the correlation was stronger for females than for males. Although women are more likely to survive an initial stroke, they have poorer functional outcomes and reduced quality of life [24]. One possible cause for this difference is a sex-specific immune response. Male mice have larger infarct volumes after a transient proximal MCAO, and this difference is eliminated with a splenectomy, suggesting that peripheral immune cells such as monocytes are affecting stroke outcomes in males [35].

In aged mice, the transcriptional response to stroke differs from that of young mice, coinciding with an increased vulnerability to stroke with age. It has been shown that the synaptic and axonal maintenance programs of aged mice are downregulated and are coupled with an overactivation of type 1 interferon signaling following ischemia [36]. These changes correlate with a slower recovery in older mice, which includes a deficiency of angiogenesis [37]. It is also known that aged blood impairs stem cell activity and activates microglia in the brain, which affects graft outcomes and could explain the poor quality of some of our grafts in aged mice [34].

In sum, our study advances the field’s understanding of what factors must be taken into account to minimize variability when performing transplantations. Nevertheless, our study is limited. Although the functional outcome is ultimately what is most important, the goal of the current study was to take early steps in identifying variables that might improve the consistency of grafting, which would allow progress toward ultimately replacing lost tissue with functional new tissue. The current state of grafts, including those described here, are still far from representing functional tissue (because of, for example, the disorganization of the cells within the graft and the lack of certain cell types, which will impair connectivity and function). Hence, an analysis of behavioral or cognitive outcomes with an expectation that the electrophysiological activity of grafts would underlie a useful behavioral improvement in the host is premature. This is also because improvements observed thus far in behavioral outcomes after grafting can be attributed to bystander effects rather than to electrophysiological activity of grafts [5,6,7]. Future improvements in cerebral grafting that include appropriate cytoarchitecture and cell compositions will be necessary to achieve the repair.

## Figures and Tables

**Figure 1 cells-11-01656-f001:**
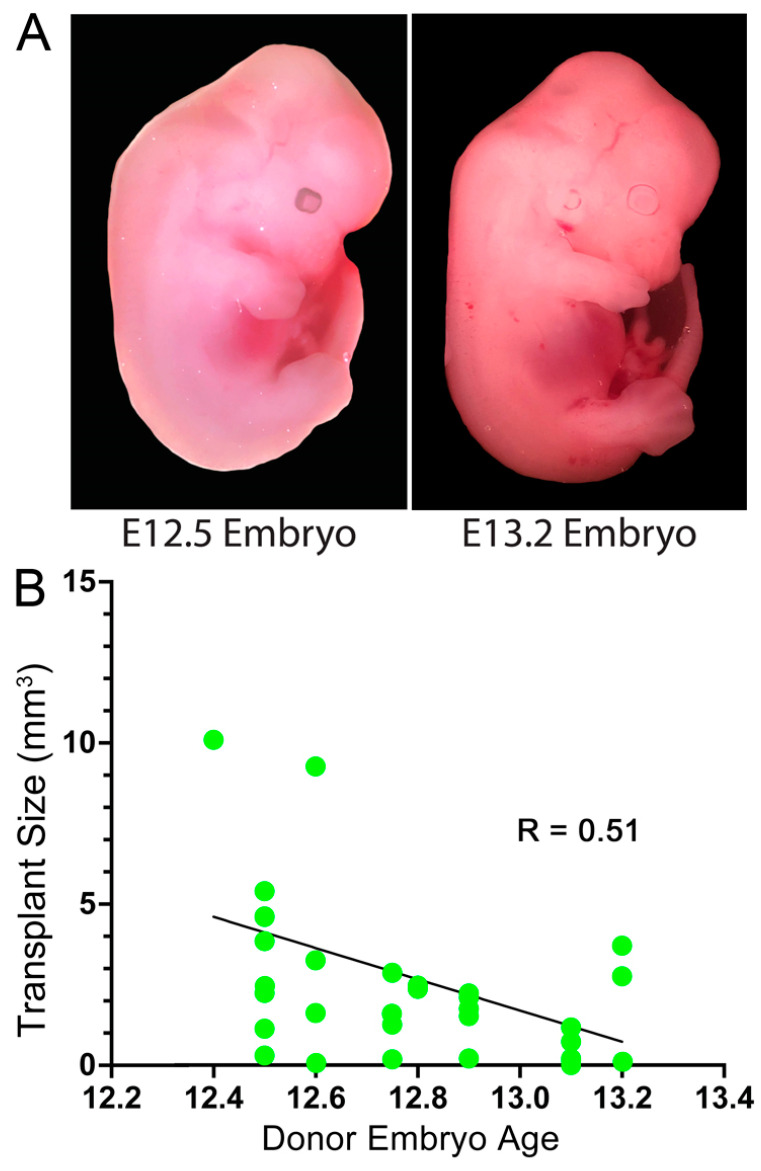
Size of transplants after 2 weeks correlated with the embryonic stage of the donors. (**A**) Example mouse embryos at the times of harvest. (**B**) Transplants within stroke sites are graphed according to the embryonic stage from which the donor cells were harvested. Each dot is a donor mouse. Spearman correlation, n = 32.

**Figure 2 cells-11-01656-f002:**
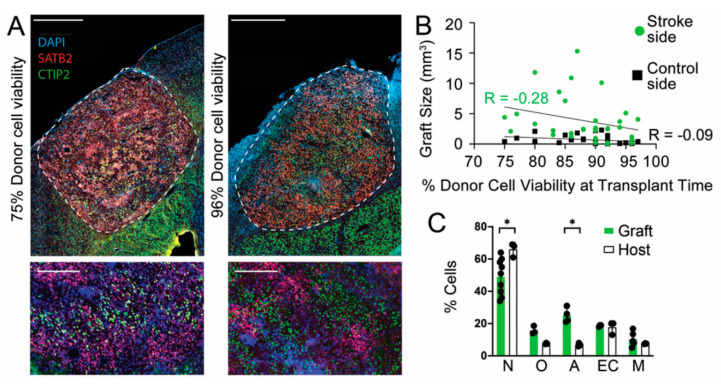
Size of transplants did not correlate with donor cell viability at the time of transplantation. (**A**) Sections of transplants at stroke sites with 75% (left) and 96% (right) donor cell viability at the time of transplant. Scale bars: upper panels, 500 µm; lower panels, 200 µm. (**B**) Transplant size is graphed according to the percent of viable cells immediately before transplantation into stroke (green) and contralateral control (black) sites. Each dot is a host mouse. Spearman correlation, n = 30 for each. (**C**) Cell types within grafts are compared with endogenous cells. Immunostaining was performed for SATB2 and CTIP2 (neurons), OLIG2 (oligodendrocytes), GFAP (reactive astrocytes), CD105 (endothelial cells), and IBA1 (microglia). For neurons, * *p* = 0.018; for reactive astrocytes, * *p* = 0.029. Mann–Whitney test, +/− s.t.d.

**Figure 3 cells-11-01656-f003:**
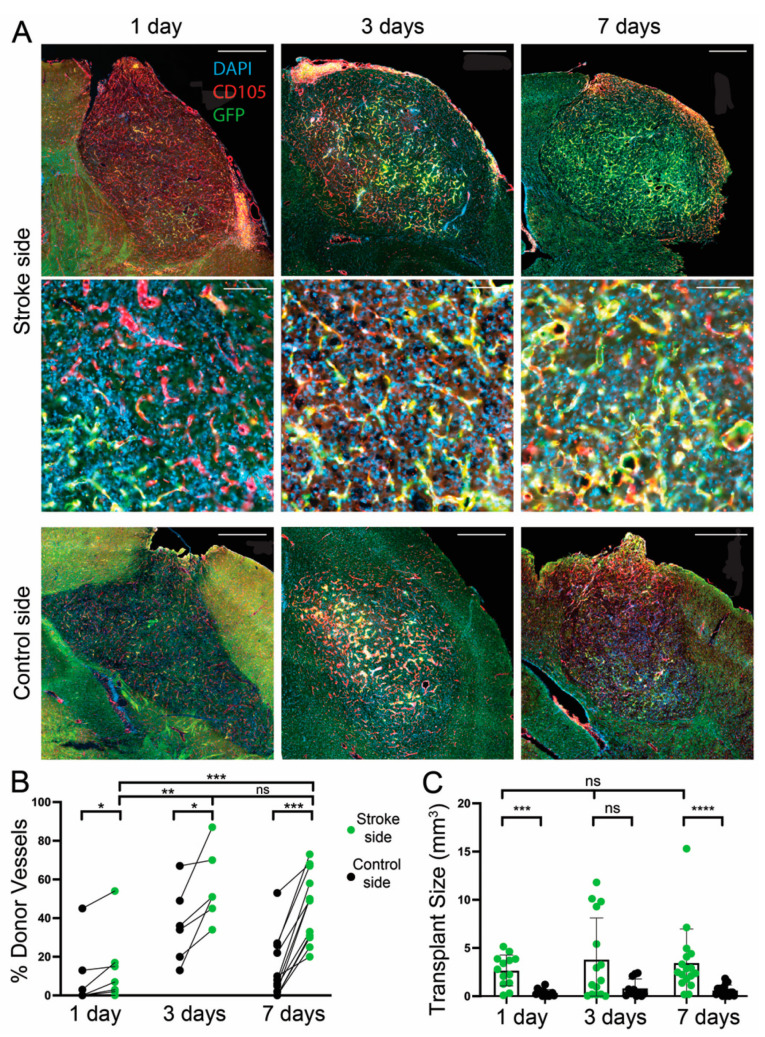
Effect of a transplantation delay of 1, 3, or 7 days after stroke. (**A**) Example images of transplants with a 1-, 3-, and 7-day delay between dMCAO and transplantation (with all brains collected 14 days after transplantation). Top—stroke side; bottom—control side of the same mouse. Scale bars: low magnification, 500 µm; higher magnification, 100 µm. (**B**) Percentage of blood vessels from the donor cell population in grafts. For 1-day delay (n = 7), * *p* = 0.03; 3-day delay (n = 6); 7-day delay (n = 12), *** *p* = 0.0005, and 1 vs. 7-day, *** *p* = 0.0005, 1 vs 3-day, ** *p* < 0.01. Each pair. Each pair is one animal; Wilcoxon Test. (**C**) Size of transplants grouped for control and stroke sides. Each dot is a host mouse. For 1-day delay (n = 13), *** *p* = 0.006; 3-day (n = 15), *p* = 0.26; 7-day (n = 17), **** *p* < 0.0001; and 1- vs. 7-day *p* = 0.4; ns, not significant. Mann–Whitney Test.

**Figure 4 cells-11-01656-f004:**
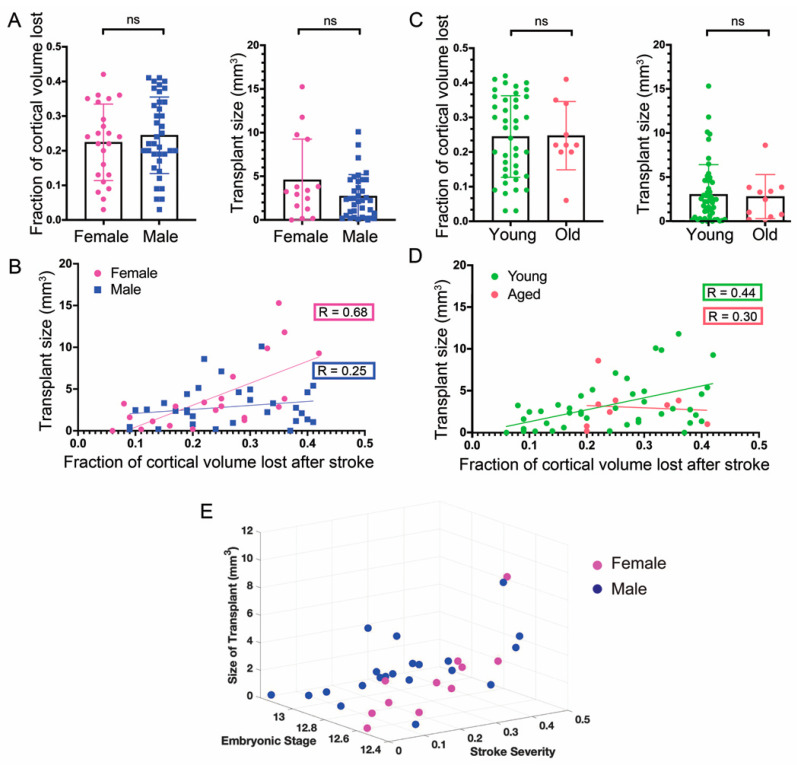
Effects of sex and age on transplants. Average stroke severity and transplant size in female versus male mice (**A**) and young versus old mice (**C**); ns, not significant. Size of transplant after 2 weeks in female versus male mice (**B**) and in young and old mice (**D**) as it correlates to the volume of cortical loss (stroke severity). Mann–Whitney Test *p* < 0.02. Spearman correlation. Each dot is a mouse. In (**A**,**B**), n = 36 males and 23 females; in (**C**,**D**), n = 45 young and 10 old mice. (**E**) A general linear model was applied to examine the correlation between graft size, the embryonic stage of donor cells, and stroke severity (see also text for details).

**Figure 5 cells-11-01656-f005:**
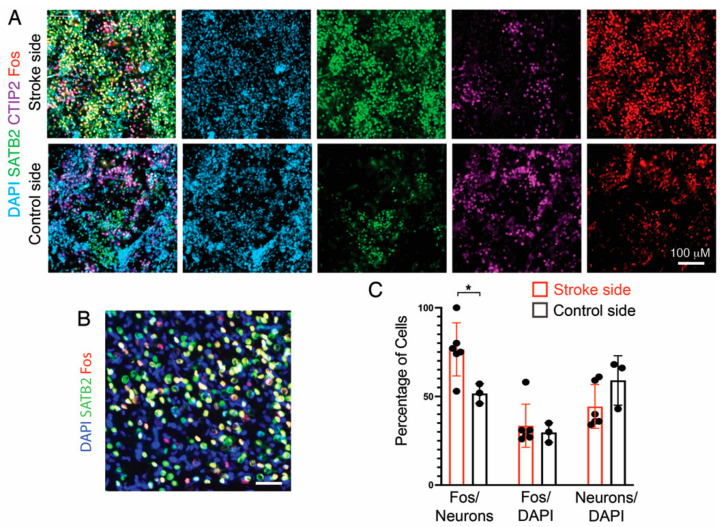
Analysis of Fos expression in grafts with and without stroke. (**A**) Representative immunofluorescent stains of grafts at stroke or control sites for SATB2 (green), CTIP2 (purple), and Fos (red). DAPI counterstain (blue). (**B**) Higher magnification view of staining of graft at a stroke site. Scale bar: 20 µM. (**C**) Quantitation of Fos+ cells. Each dot is the average from multiple sections of one mouse. Mann–Whitney test, * *p* < 0.05, +/− s.t.d.

## Data Availability

Not applicable.

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
