# Peer review of "Potential Variables for Improved Reproducibility of Neuronal Cell Grafts at Stroke Sites"

_cells, 2022, doi:10.3390/cells11101656_

Round 1

Reviewer 1 Report

The authors have responded the questions but only by explanation rather than performing more experiments. 

Author Response

Reviewer 1 comments: The authors have responded the questions but only by explanation rather than performing more experiments.

Authors' response: In the previous revision, new Figure 5 was added describing Fos expression as a proxy for neuronal activity in the graft. We have now also added new Figure 2C describing the cell composition of the grafts compared with neighboring host tissue. The appropriateness of the research design in now discussed in a new paragraph describing the limitations of this study (see also response to Reviewer 3 point 1). We have made additions to the Methods regarding donor cells and host animals (see also response to Reviewer 2’s point 1 and Reviewer 3’s point 3).

Reviewer 2 Report

Dear Authors,

The manuscript entitled " Potential variables when grafting cells for stroke repair" is a very well prepared and written article. The language that has been used from the authors is very good. Both introduction and discussion section are containing valuable information, promoting further the content of the study.

Only some minor suggestions i have for the submitted article.

1) In materials and methods, section Preparation of Donor Cells from Mouse Embryos and Transplantation, the authors indicated the cell isolation that were further used for the transplantation experiments. 

The authors needs to clarify what kind of these stem cells are, and furthermore characterization of the stem cell population must be performed.

2) The title of the article needs revision, in order to be focused on the content of the manuscript.

Author Response

Reviewer 2 comments: Only some minor suggestions i have for the submitted article.

1) In materials and methods, section Preparation of Donor Cells from Mouse Embryos and Transplantation, the authors indicated the cell isolation that were further used for the transplantation experiments. 

The authors needs to clarify what kind of these stem cells are, and furthermore characterization of the stem cell population must be performed. 

2) The title of the article needs revision, in order to be focused on the content of the manuscript.

Authors' response: 1) The composition of the donor cell population has previously been described. To make this clear, we have added a sentence and a citation in the Methods sections describing donor cells, which include neuronal lineage cells, vascular cells, and microglia.

2) We have revised the title accordingly.

Reviewer 3 Report

I have a few minor comments:
1.The study has some limitations due to the type of experiment carried out, it should be clearly indicated through the paragraph "study limitations"
2. In the introduction, the authors indicated that we do not currently have very effective methods of treating stroke etc - there is no reference to this part at the end of the article - the authors' results are promising, but what is their clinical impact on stroke patients?
3. the materials and methods do not explicitly state the size of the group of animals that were used for the study

Author Response

Reviewer 3 comments: I have a few minor comments:
1. The study has some limitations due to the type of experiment carried out, it should be clearly indicated through the paragraph "study limitations"
2. In the introduction, the authors indicated that we do not currently have very effective methods of treating stroke etc - there is no reference to this part at the end of the article - the authors' results are promising, but what is their clinical impact on stroke patients?
3. the materials and methods do not explicitly state the size of the group of animals that were used for the study. 

Authors' response: 1. We have added a paragraph at the end of the Discussion specifically describing the limitations of this study.

2. To clarify this point, in the new paragraph at the end of the Discussion, we make it clear that our approach is only part of initial steps toward being able to repair neocortical tissue after stroke damage, and we do not claim to have yet succeeded.

3. The total number of animals is now included in the Methods section “Animals”.

Round 2

Reviewer 1 Report

The authors have answered the questions.

This manuscript is a resubmission of an earlier submission. The following is a list of the peer review reports and author responses from that submission.

Round 1

Reviewer 1 Report

The manuscript, entitled "Potential Variables when Grafting Cells for Stroke Repair" by Krzyspiak et al tested the different conditions about stem cell therapy in a rodent model of acute ischemic stroke. They concluded that graft size at stroke sites correlates with size of ischemic damage, host sex, and host age, but not with time of transplant after stroke. Some problems are listed below:

  1. This study investigated the effects of different conditions in stem cell therapy and confirmed that certain factors will influence the graft size after transplantation. Although they claimed that their findings can serve as a guide to improve reproducibility of cell-based repair therapies, the study outcome (transplant size) is not an useful marker for future clinical applications. Using functional outcome and/or infarct size will be more clinical relevant.
  2. In this study, the transplantation timing and age of mice did not influence the final transplant size. However, equal transplant size does not guarantee similar functional outcome. For future clinical trials, to find out an optimal treatment condition with the best functional outcome is more important.
  3. This study is essentially an association study without mechanistic investigation.
  4. While stroke includes both ischemic stroke and hemorrhagic stroke, the authors only investigate the field of ischemic stroke but use the term "stroke" throughout the manuscript. The term of ischemic stroke will be more appropriate.
  5. In Introduction, the description of "the only treatment for stroke is the injection of a thrombolytic to reperfuse the ischemic tissue" is not true that endovascular thrombectomy is another effective treatment for ischemic stroke.
  6. In Introduction, the description of "In an attempt to improve neurological function, stem cell transplantation therapy has shown promise in improving functional outcomes in clinical trials with minimal adverse events." requires some references.

Author Response

1. This study investigated the effects of different conditions in stem cell therapy and confirmed that certain factors will influence the graft size after transplantation. Although they claimed that their findings can serve as a guide to improve reproducibility of cell-based repair therapies, the study outcome (transplant size) is not an useful marker for future clinical applications. Using  functional outcome and/or infarct size will be more clinical relevant. 

Response: We agree with the reviewer that size does not necessarily relate to the functional outcome of the transplant. However, due to the outstanding bottleneck that the field faces with cell survival, our aim in this study is to optimize transplant size and survival to set up a platform for subsequent successful functional experiments. In addition, although cell type differentiation and electrophysiological integration, for example, would be superior measures of graft functionality, these were previously examined and found not noticeably vary with the observed variations in graft size using the same experimental paradigm (Krzyspiak et al., 2022), suggesting that size in this case can serve as a proxy for levels of functionality similar to those obtained previously. Text to this effect has been added to the first paragraph of the Discussion. 

2. In this study, the transplantation timing and age of mice did not influence the final transplant size. However, equal transplant size does not guarantee similar functional outcome. For future clinical trials, to find out an optimal treatment condition with the best functional outcome is more important. 

Response: We agree, functional outcome is most important. As indicated, the goal of the current study is to take early steps in identifying variables that might improve the consistency of grafting, which would allow progress toward ultimately replacing lost tissue with functional new tissue. We believe the current state of grafts, including the grafts in our current paradigm, are still a far cry from representing functional tissue (due for example to the disorganization of the cells within the graft, which we know will impair optimal connectivity). Hence, an analysis of outcomes with an expectation that electrophysiological activity of grafts underlies a useful behavioral improvement to the host, we believe is at this stage premature. This is also because we know that improvements observed thus far in behavioral outcomes after grafting, rather than due to electrophysiological activity of grafts (i.e. true repair), can instead be attributed to bystander effects (as described in the text), which are not the focus of this study. 

3. This study is essentially an association study without mechanistic investigation. 

Response: Because the steps we are taking are too early to expect true tissue repair, assessing mechanisms by which stroke outcomes might improve would be premature (and would in any case likely reflect bystander effects rather than repair). Please also see Responses to points 1 and 2 above. 

4. While stroke includes both ischemic stroke and hemorrhagic stroke, the authors only investigate the field of ischemic stroke but use the term "stroke" throughout the manuscript. The term of ischemic stroke will be more appropriate. 

Response: Thank you, we have replaced “stroke” with “ischemic stroke” at key places throughout the manuscript. 

5. In Introduction, the description of "the only treatment for stroke is the injection of a thrombolytic to reperfuse the ischemic tissue" is not true that endovascular thrombectomy is another effective treatment for ischemic stroke. 

Response: Thank you, this has been corrected in the first paragraph of the Introduction. 

6. In Introduction, the description of "In an attempt to improve neurological function, stem cell transplantation therapy has shown promise in improving functional outcomes in clinical trials with minimal adverse events." requires some references. 

Response: Thank you, references have been added to the second paragraph of the Introduction. 

Reviewer 2 Report

The authors propose a method for using cell grafts from ischemic tissue repair and the variables associated with it. Some suggested edits are listed below.

Introduction:

-Mechanical thrombectomy is another treatment method other than intravenous therapy for ischemic stroke treatment. This should be mentioned early in the introduction or the first paragraph should be reworded to state "therapy."

Results:

-Was there any thought to do the transplant beyond the 7 day mark as sometimes individuals are "stubborn" and wait a couple weeks before having their symptoms diagnosed? It would be interesting to see if beyond that 7 day make lead to promising results.

Author Response

Introduction:  

-Mechanical thrombectomy is another treatment method other than intravenous therapy for ischemic stroke treatment. This should be mentioned early in the introduction or the first paragraph should be reworded to state "therapy." 

Response: Thank you, this has been corrected in the first paragraph of the Introduction. 

Results:

-Was there any thought to do the transplant beyond the 7 day mark as sometimes individuals are "stubborn" and wait a couple weeks before having their symptoms diagnosed? It would be interesting to see if beyond that 7 day make lead to promising results. 

Response: We agree transplanting with a significant delay relative to the injury would be desirable in the clinic. However, current studies show that once an injury has healed, the environment is much less likely to integrate new neurons efficiently. Other studies have looked past 7 days, and found that 7 days is superior, therefore our interest was to determine if earlier timepoints would be even better due to the rapidly changing environment of the injured tissue. In the future, we will also explore whether reactivating the injury response at later times, for example by removing scar tissue, might also favor engraftment. 

Reviewer 3 Report

Manuscript by Krzyspiak et al. explores different factors such as host sex age, extent of ischemic damage, and time of transplant after ischemia as possible factors affecting engraftment of donor cell in stroke lesioned cortex.

The major weakness of the manuscript that it explores the effect gf the above factors on the size of the graft and not the neurons and other cells generated from the grafted cells. There is no attempt to describe or evaluate the cellular composition and possible differential effect of different factors on this composition, selective survival and differentiation potential.

Another major methodological weakness is that authors do not represent numbers of animals in each experimental group and do not show and describe statistical analysis.

And, last but not least major weakness is the source of the cells – fetal tissue which lacks any translational value and exploring logistical details of this source for cell therapy of ischemic stroke does not seem to have any future clinical use.

Specific comments:

  1. Authors are forgotten g that thrombectomy is also important procedure to treat ischemic stroke acutely.
  2. Figure 1. There is not a strong correlation between transplant size and age of the donor ( Number or donors should be increased to make this statement. One of the data is 0 (transplantation didn't work?).
  3. It is not clear how was the success ratio of transplantation? (Number of animals transplanted vs. Number of animals with transplantation afterwards).
  4. Figure 2A. Higher magnification of the images should be shown. Cortical markers are not visible. It would be important to show and discuss the expression pattern of deep and upper cortical markers. Same for Figure 3A, higher magnification as well as individual staining is needed.
  5. Results have too many references to previous published studies making difficult to distinguish between new data and published data. Discussion of the results should be done in the Discussion section.
  6. Figure 4. State number of animals per group. Number of male animal when quantifying transplant size looks much higher that male animals used for quantifying fraction of cortical volume loss, why are not all the animals included in this quantification? In the groups young and old mice, is there a comparable contribution of female and male animals?
  7. An in vitro characterization of the cells to be transplanted and the quantification of the infarct volume should be presented.

Author Response

The major weakness of the manuscript that it explores the effect gf the above factors on the size of the graft and not the neurons and other cells generated from the grafted cells. There is no attempt to describe or evaluate the cellular composition and possible differential effect of different factors on this composition, selective survival and differentiation potential.

Response: We agree with the reviewer that size does not necessarily relate to the functional outcome of the transplant. However, due to the outstanding bottleneck that the field faces with cell survival, our aim in this study is to optimize transplant size and survival to set up a platform for subsequent successful functional experiments. In addition, although cell type differentiation and electrophysiological integration, for example, would be superior measures of graft functionality, these were previously examined and found not noticeably vary with the observed variations in graft size using the same experimental paradigm (Krzyspiak et al., 2022), suggesting that size in this case can serve as a proxy for levels of functionality similar to those obtained previously. Text to this effect has been added to the first paragraph of the Discussion. 

Another major methodological weakness is that authors do not represent numbers of animals in each experimental group and do not show and describe statistical analysis. 

Response: Each dot in each graph represents 1 animal. We have now also added the numbers for N’s to the text and in each figure legend, and included a section in Methods on Statistical Analyses. 

And, last but not least major weakness is the source of the cells – fetal tissue which lacks any translational value and exploring logistical details of this source for cell therapy of ischemic stroke does not seem to have any future clinical use. 

Response: We agree that fetal tissue is not a good source for use in clinics (even though fetal tissue has been used in patients for treating Parkinsons, and could in principle also be used for stroke). Nevertheless, using dissociated cells from fetal neocortical tissue, for which the cell types have been well described, provides a benchmark for developing similar cell populations derived from human ES or iPS cells for clinical translation. 

Specific comments: 

1. Authors are forgotten g that thrombectomy is also important procedure to treat ischemic stroke acutely. 

Response: Thank you, this has been corrected in the first paragraph of the Introduction. 

2. Figure 1. There is not a strong correlation between transplant size and age of the donor ( Number or donors should be increased to make this statement. One of the data is 0 (transplantation didn't work?).

Response: We have now added more N’s. Indeed, a value of 0 or near 0 indicates no or little graft tissue present. 

3. It is not clear how was the success ratio of transplantation? (Number of animals transplanted vs. Number of animals with transplantation afterwards).

Response: All animals that received transplanted cells were included in the study. There were no exclusion criteria. If there was no or little transplant detected, this was reported (as 0 or near 0, respectively) on the axis of the graphs for size, examples of which can be found in all Figures. 

4. Figure 2A. Higher magnification of the images should be shown. Cortical markers are not visible. It would be important to show and discuss the expression pattern of deep and upper cortical markers. Same for Figure 3A, higher magnification as well as individual staining is needed. 

Response: Thank you, we have now added higher magnification images to Figures 2 and 3. Please note that we have also addressed this question in depth in a previous study, Krzyspiak et al. “Donor-derived vasculature is required to support neocortical cell grafts after stroke.” Stem Cell Research Volume 59, 2022  doi: 10.1016/j.scr.2021.102642. 

5. Results have too many references to previous published studies making difficult to distinguish between new data and published data. Discussion of the results should be done in the Discussion section. 

Response: The references in the Results provide the rationales for our experiments and explain our choices of experimental approaches. If moved to the Discussion, the text would follow a less coherent flow. Nevertheless, for increased clarity, we have now added “in a previous study” or similar where appropriate. 

6. Figure 4. State number of animals per group. Number of male animal when quantifying transplant size looks much higher that male animals used for quantifying fraction of cortical volume loss, why are not all the animals included in this quantification? In the groups young and old mice, is there a comparable contribution of female and male animals? 

Response: Thank you, the number of animals are now indicated in the text and legends.    Significant ischemic damage failed to occur more often in females after dMCAO, so N is smaller for females. 

7. An in vitro characterization of the cells to be transplanted and the quantification of the infarct volume should be presented. 

Response: Thank you, the method for quantifying infarct volume has now been added to the “Induction of Distal Medial Cerebral Artery Occlusion (dMCAO)” section of the Materials and Methods. The cells used for transplantation, E12.5 mouse neocortex, are comprised of neural, vascular, and microglial cells (Loo et al., 2019,    Single-cell transcriptomic analysis of mouse neocortical development. Nat Commun. 10:134). 

Round 2

Reviewer 1 Report

The authors have already performed some minor modification of the manuscripts. But extra experiments are lacking using functional outcome to evaluate the effects of stem cells. 

Reviewer 3 Report

Authors failed to deal with major concerns which remain after revision.